# WISE-GNN: ENHANCING GNNS WITH WISE EMBEDDING AND TOPOLOGICAL ENCODING

## ABSTRACT

Graph Neural Networks (GNNs) have emerged as a powerful framework for graph representation learning. However, they often struggle to capture long-range dependencies between distant nodes, leading to suboptimal performance in tasks such as node classification, particularly in heterophilic graphs. Challenges like oversmoothing, oversquashing, and underreaching intensify the problem, limiting GNN effectiveness in such settings.

In this paper, we introduce *WISE-GNN*, a novel framework designed to address these limitations. Our approach enhances any GNN model by incorporating *Wise-embeddings*, which capture attribute proximity and similarities among distant nodes, thereby improving the representation of nodes in both homophilic and heterophilic graphs. Additionally, we propose a topological module that can be smoothly integrated into any GNN model, further enriching node representations by incorporating the topological signatures of node neighborhoods. Comprehensive experiments across various GNN architectures show that WISE-GNN delivers significant improvements in node classification tasks, achieving mean accuracy gains of up to 14% and 23% on benchmark datasets in homophilic and heterophilic settings, respectively. Moreover, WISE-GNN enhances the performance of various GNN architectures, allowing even standard GNNs to outperform SOTA baselines on benchmark datasets.

## 1 INTRODUCTION

GNNs have emerged as the primary approach for graph representation learning over the last decade. They have exhibited remarkable performance across various tasks such as node classification, link prediction, and graph classification (WPC+20; MCT+24). The fundamental concept involves learning node representations by integrating both structural and node attribute data through the aggregation of messages from neighboring nodes. Despite their achievements, recent research highlights potential drawbacks associated with GNNs, including issues like oversmoothing, oversquashing, and underreaching, which can vary depending on the graph's structure or the architecture of the GNN itself (LBYS23).

The oversquashing issue in GNNs arises when node representations lose sensitivity to information originating from crucial yet distant nodes in the network (TDGC+21; DGGB+23). Conversely, the underreaching problem manifests as an inability to fully explore or influence all pertinent nodes within the graph, leading to information degradation (BKM+20). Despite their spatial distance in the graph, nodes may exhibit proximal representations in attribute space, intensifying the oversquashing and underreaching problems, particularly impeding message-passing GNNs in capturing insights from remote but relevant nodes. This deficiency, especially pronounced in heterophilic settings, detrimentally impacts tasks such as node classification, underscoring the importance of addressing these challenges.

While many GNNs leverage node attributes as initial embeddings to infuse this critical information into the model through message aggregation, the dimension of the latent space where node embeddings reside has a profound effect on the learning of GNNs (DJ23). Furthermore, in scenarios where node attributes are unavailable, several conventional methods are employed to initialize node embeddings, thereby implicitly determining the dimensionality of the latent space. These choices, coupled with the mentioned problems, inevitably lead to suboptimal performance of GNNs.

To mitigate these challenges, we introduce a novel positional encoding method called *Wise embedding*, which captures attribute proximity within node representations and directly integrates it into GNN embeddings. The core idea is to encode the position of node feature vectors in the feature space by capturing their relative positions with respect to our *classWISE landmarks*. This method aims to improve GNNs by incorporating missing information from relevant but distant nodes, leading to more robust node representations. Furthermore, the local subgraph structure is pivotal in enhancing GNN performance, with several models integrating subgraph information to improve expressiveness and capture finer structural details. For example, Barcelo et al. (BKM[+]20) introduce local subgraph parameters within GNNs to better capture structural nuances. Subgraph-level encoding is explored in G-meta (HZ20), which applies GNNs on rooted subgraphs for meta-learning purposes. Similarly, k-hop GNNs (NDV20) and Ego-GNNs (SVH21) utilize rooted subgraphs and sequential message passing to encode local substructures. ID-GNNs (YGSYL21) enhance message passing by incorporating node identities through k-egonets, leading to improved structural representations. However, many of these methods face scalability issues due to the combinatorial complexity of subgraph computations, such as solving subgraph isomorphisms (TZK21). Our approach addresses these limitations by using persistent homology to capture critical subgraph structures within the graph. Persistent homology offers a powerful topological framework for encoding multi-scale subgraph information. This method not only captures intricate topological features but also maintains scalability, making it suitable for large-scale graph tasks while enhancing the overall performance of GNNs. By augmenting the GNN framework with attribute-aware Wise embeddings, and exploring local subgraph structures via persistent homology, our model significantly improves the ability of GNNs to recognize and utilize both structural and attribute proximities, thereby enhancing their performance across various graph-based tasks.

Our contributions can be summarised as follows:

- We propose WISE-GNNs, a novel enhancement of GNNs that effectively integrates node attribute proximity information to enrich node representation learning.

- We introduce Wise-embeddings, which measures node similarity relative to individual classes, effectively capturing a node's positional relationships within the attribute space.

- By initializing GNNs with Wise-embeddings, we pre-inform the model of relational structures in the attribute space, enabling them to capture long-range dependencies. Wise embeddings are model-agnostic and can be easily integrated into any GNN architecture with minimal code adaptation.

- To improve robustness, we integrate topological structure information from node neighborhoods, enhancing the model's ability to capture both local and global graph properties.

- Extensive experiments across multiple GNN architectures show that integrating Wise-embeddings significantly enhances node classification performance, achieving up to a 14% and 23% increase in average accuracy gains on benchmark datasets in homophilic and heterophilic settings, respectively.

## 2 BACKGROUND

### 2.1 GRAPH NEURAL NETWORKS

GNNs are neural networks designed to process graph-structured data. In the predominant message-passing scheme, GNNs update node embeddings through information from neighboring nodes. The versatility of GNNs has led to their widespread use across domains such as social networks, recommendation systems, drug discovery, and knowledge graphs. Notable variants include GCNs (KW17), GATs (VCC[+]18), GraphSAGE (HYL17), and GINs (XHLJ19), each suited for different graph types and tasks like node classification, graph classification, and link prediction. Despite their success, GNNs face challenges, including loss of structural information and difficulty in capturing long-range dependencies and multiscale information (PWC[+]19; LHKX22; LHL[+]22).

Several recent studies address these issues. MixHop (AEHPK[+]19) extracts features from multi-hop neighborhoods to enhance information retrieval, while (YJJ[+]20) introduces metrics for heterophilic graphs. Geom-GCN (PWC[+]19) preserves structural information through bi-level aggregation, and FAGCN (BWSS21) enhances GAT by integrating edge-level aggregation for high-frequency signals.

GPRGNN (CPLM20) introduces adaptive weights for heterophilic graphs. However, no prior work has systematically analyzed attribute proximity. For recent advances, refer to surveys (WPC$^+$20; XWDG22; ZZH$^+$22).

## 2.2 GNNs and Heterophily

A major challenge with most GNNs is their reliance on the homophily assumption, which means that neighboring nodes tend to have similar labels or features. However, in many real-world networks, such as protein interaction and web networks, this assumption does not hold, as connected nodes often have different features or labels (PWC$^+$19; ZYH$^+$23). In these heterophilic networks, traditional GNN models may perform poorly, sometimes even worse than simpler models like multilayer perceptrons (ZYZ$^+$20; LHL$^+$22).

To address this issue, recent research has focused on developing GNNs that work better in heterophilic environments. These efforts can be grouped into two main approaches. The first approach aims to make the input features more informative for the GNN (PWC$^+$19; XDZW22; XCZ$^+$23), while the second approach focuses on improving how information is passed and aggregated between nodes to better suit heterophilic networks (HWX$^+$21; YLL$^+$21; LHL$^+$21; LHX$^+$23a).

Moreover, recent studies have analyzed heterophily from different angles and proposed new methods to tackle these challenges in graph representation learning (ZLP$^+$22; LHX$^+$23b; MCJ$^+$24; RCDG$^+$24). These advancements are helping to build GNN models that can handle both homophilic and heterophilic networks more effectively.

## 2.3 Persistent Homology

To capture the deeper, often hidden, structural properties of node neighborhoods, we employ Persistent Homology (PH), a key technique in Topological Data Analysis (TDA). Unlike conventional methods, which may focus solely on graph or metric properties, PH provides a powerful tool to quantify topological features—such as clusters, loops, and voids—that persist across multiple scales. By examining how these features evolve and persist, PH reveals intricate patterns in the data that might otherwise remain undetected. This approach offers broader applicability beyond graphs, extending to point clouds, images, and other complex datasets. In the context of graphs, PH helps us uncover latent topological insights that complement traditional graph-based analysis. For a deeper dive into PH across different data types, refer to (DW22; CA24).

We can summarize PH as a three-step process. Let $\mathcal{G} = (\mathcal{V}, \mathbb{E})$ be a graph with a node set $\mathcal{V}$ and an edge set $\mathbb{E}$. The first step is called *filtration*, where we construct a nested sequence of simplicial complexes induced from the graph. A common method is to get a nested sequence of subgraphs $\mathcal{G}^1 \subseteq \ldots \subseteq \mathcal{G}^N = \mathcal{G}$. A common approach involves employing a filtration function $f : \mathcal{V} \to \mathbb{R}$ alongside a set of thresholds $\mathcal{I} = \{\epsilon_i\}$, where $\epsilon_1 = \min_{v \in V} f(v) < \epsilon_2 < \ldots < \epsilon_N = \max_{v \in \mathcal{V}} f(v)$. For each $\epsilon_i$ in $\mathcal{I}$, a subset $\mathcal{V}_i = \{v_r \in \mathcal{V} \mid f(v_r) \leq \epsilon_i\}$ is formed. Then, the induced subgraph $\mathcal{G}^i$ by $\mathcal{V}_i$, denoted as $\mathcal{G}^i = (\mathcal{V}_i, \mathcal{E}_i)$, is constructed, where $\mathcal{E}_i = \{e_{rs} \in \mathbb{E} \mid v_r, v_s \in \mathcal{V}_i\}$. Next, we obtain a simplicial complex for each subgraph, yielding a *filtration* $\widehat{\mathcal{G}}^1 \subseteq \ldots \subseteq \widehat{\mathcal{G}}^N$. A common method is to use clique complexes, where the clique complex $\widehat{\mathcal{G}}_i$ of $\mathcal{G}_i$ is obtained by adding $k$-simplices for every complete $(k+1)$-subgraph in $\mathcal{G}_i$. The common filtration functions are degree, betweenness, centrality, or an attribute-specific function. Similarly, one can employ edge weights (or Ricci functions) to create an edge filtration from the graph (AAEF19).

The second step extracts *persistence diagrams*, where we systematically trace the evolution of topological features in the filtration $\{\widehat{\mathcal{G}}_i\}$. These topological features, such as connected components (0-holes), loops (1-holes), and cavities (2-holes), are represented by $k$-dimensional topological features or $k$-holes. By using homology groups (Hat02), PH meticulously records the appearance and disappearance of these features in the filtration, and records their birth and death times in persistence diagrams. In particular, the $k^{th}$ persistence diagram $\mathrm{PD}_k(\mathcal{G}) = \{(b_\sigma, d_\sigma) \mid \sigma \in H_k(\widehat{\mathcal{G}}^i)$ for $b_\sigma \leq i < d_\sigma\}$ where $H_k(\widehat{\mathcal{G}}^i)$ represents the $k^{th}$ homology group of $\widehat{\mathcal{G}}^i$, consists of pairs $(b_\sigma, d_\sigma)$ for each $k$-hole $\sigma$, where $b_\sigma$ and $d_\sigma$ denote the birth and death times, respectively.

The final step is the *vectorization* process. While PH extracts hidden shape patterns from data in the form of persistence diagrams (PDs), these PDs, consisting of points (birth times, death times)

in $\mathbf{R}^2$, are not inherently practical for machine learning (ML) tasks. Instead, common techniques involve faithfully representing PDs as kernels (KJM20) or vectorizations (AAJ$^+$23). Vectorizations transform the obtained PDs into a function or vector format, making them more suitable for ML tools. Common vectorization methods include Persistence Images, Persistence Landscapes, Silhouettes, and various Persistence Curves, including Betti curves (AAJ$^+$23). While these methods offer flexibility in model and data analysis, it is also common to use automated neural network approaches to avoid the need for manual vectorization choices or hyperparameter tuning (CCI$^+$20; HKN19).

## 3 METHODOLOGY

**Motivation.** Our objective is to address *oversquashing* and *underreaching* by incorporating key information from distant nodes early in the learning process. These challenges stem from the GNN's failure to access crucial information from highly relevant yet distant nodes. Drawing an analogy between networks and societies, individuals may reside far apart but have very close interests. Unfortunately, the message-passing algorithm of GNNs only considers messages from nearby nodes, resulting in the omission of vital information derived from the attribute proximity of nodes (shared interests) necessary for improved node representation.

**Solution.** To overcome these limitations, we introduce node representations called *Wise Embeddings*, designed to capture the missing attribute similarity information for both nearby and distant nodes. By integrating this essential information early in the learning process, Wise Embeddings act as a "vaccine" for GNNs, shielding them from the issues outlined earlier. To further enhance the robustness of node representations, we also incorporate topological information from node neighborhoods through a topological module. As depicted in Figure 1, our framework uniquely combines these elements to create effective and meaningful node representations.

### 3.1 WISE EMBEDDDINGS

In a graph, a node's spatial information can be effectively represented by its neighboring nodes. However, when it comes to capturing its features (interests), the process is not as straightforward. Our objective is to introduce *classWISE landmarks* and utilize these landmarks to position the node's feature vector within the feature space. This involves establishing a "coordinate system" by measuring the distances between the node's feature vector and the landmarks.

Node features vary across different types of networks. In social networks, they typically represent user-specific details, whereas in citation networks, they often indicate the presence or absence of particular keywords. While the structural interactions among neighboring nodes are essential for predicting node behavior, incorporating the inherent features of nodes into the analysis is equally important. Given that GNNs aggregate messages from nearby neighborhoods for updates, their performance heavily depends on the informativeness of the initial embedding. Moreover, as embeddings are updated by collecting information only from the local neighborhood, data from distant nodes cannot directly contribute to the process.

To address the issue, we first consider the node embeddings in the feature space. Each node $u$ is represented as a feature vector $\mathcal{X}(u)$ in the feature space $\mathbb{R}^n$, where $\mathcal{X} : \mathcal{V} \to \mathbb{R}^n$ represents a node embedding map. For each class $\mathcal{C}_j$, we form a point cluster $\mathcal{W}_j = \{\mathcal{X}(u)|u \in \mathcal{C}_j\}$ and define a *classWISE landmark* $\xi_j$ for it. Depending on the context, we can define more than one type landmark for each class $\mathcal{C}_j$, i.e., $\{\xi_j^1, \xi_j^2, \ldots, \xi_j^m\}$.

One natural method for real-valued feature vectors is averaging the feature vectors, which basically correspond to the centroid of the cluster $\mathcal{W}_j$. For binary or categorical vectors, it can be a class representative vector, e.g., the features that exist in all (or at least some percentage) of class members (selective) or the features that belong to at least one member of the class member (inclusive) (See Appendix B for examples). After defining the *classWISE landmarks* $\xi_j$ for each class $\mathcal{C}_j$, depending on the context, we define the distance $d(.,.)$ in the latent space $\mathbb{R}^n$ as a similarity measure in the feature space to measure the similarity/distance of the feature vectors to class landmarks. For real-valued vectors, Euclidean distance or cosine distance are the most natural choices. For binary vectors, Jaccard Similarity, Positive Similarity, and Cosine similarity are the most common methods.

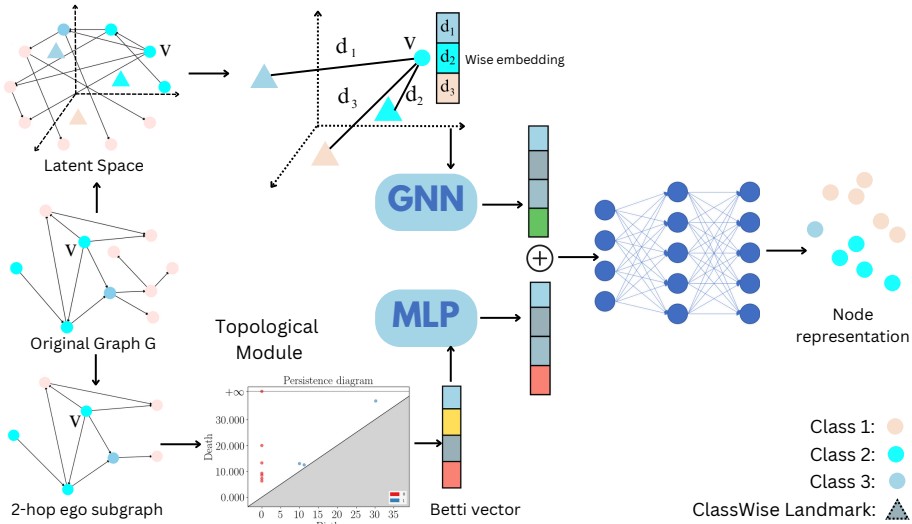

Figure 1: **WISE-GNN Flowchart**. For a given graph, we first compute Wise Embeddings in the attribute space, and use them as initial embeddings for GNNs. In the second step, we learn the topological structure of node neighborhoods and use them to enhance node representations through a neural network.

For categorical features, we utilize Positive Similarity, while for real-valued features, we employ Euclidean Distance as the similarity measure. A comprehensive description of the similarity measure is provided in Section 4.4 and Appendix A. After defining class landmarks and similarity measures, we are ready to define *Wise Embeddings* of the nodes. For a given node $u \in \mathcal{V}$, let $\mathcal{X}(u)$ be the node feature vector. For $1 \le i \le m$ (the number of landmark types), with class landmarks $\{\xi_j^i\}_{j=1}^N$, we define an $N$-dimensional vector ($N$ is the number of classes) as follows: $\vec{\alpha}^i(u) = [d_1^i, d_2^i, d_3^i, \cdots, d_N^i]$ where $d_j^i = \mathbf{d}(\mathcal{X}(u), \xi_j^i)$ for each class $\mathcal{C}_j$ where $\mathbf{d}$ is the similarity measure. For each landmark type, we have a different "coordinate system" and a different $N$ dimensional embedding of the node. We call a collection of such positional vectors $\{\vec{\alpha}^i(u)\}$ representing node feature vectors in feature space as *Wise Embeddings*.

### 3.2 TOPOLOGICAL MODULE

In this part, we aim to capture the topological structure of the neighborhoods of each node by utilizing persistent homology (Section 2.3). This process begins by constructing a k-hop ego graph from a given graph $\mathcal{G}$, where we specifically employ a 2-hop ego graph $\mathcal{G}_u$ for each node $u$ in our study considering computational complexity and performance accuracy. Then, we employ sublevel filtration, utilizing the degree function (defined in the original graph, not in the ego graphs), to derive a persistent diagram from this subgraph $\mathcal{G}_u$, effectively capturing the topological characteristics of the node neighborhood. This persistence diagram serves as the foundation for extracting local topological information, facilitated by the Betti vectors. Leveraging the Betti vectors $\vec{\beta}_0(u)$ and $\vec{\beta}_1(u)$ (subscripts representing the topological dimensions), we obtain a topological encoding that encapsulates essential structural characteristics of the graph. Finally, we integrate this topological information into our GNN architecture via a 3-layer Multi-Layer Perceptron, enhancing the GNN's capacity to understand and utilize the underlying graph topology for improved performance for downstream tasks. For a better understanding of the effects of 1, 2, and 3-hop ego graphs, we provide detailed experimental descriptions in Section 4.4.

### 3.3 WISE-GNN MODEL

With WISE-GNN framework, we can enhance any GNN by integrating positional encoding of nodes and local topological information. Initially, for a given graph $\mathcal{G}$, we use the concatenation of Wise Embeddings (Section 3.1), i.e., $\widehat{\alpha}(u) = \vec{\alpha}^1(u)\|\vec{\alpha}^2(u)\|\dots\|\vec{\alpha}^k(u)$, as the initial embedding of the node $u$ in our GNN framework, i.e. $\mathbf{h}_u^0 = \widehat{\alpha}(u)$. These initial embeddings are then updated

through the GNN, refining through the positional information of $u$ in $\mathcal{G}$. In this process, we utilize $w$-dimensional readouts to capture global positional context. Simultaneously, we extract topological features from each node's k-hop neighborhood, as outlined in Section 3.2. These features are processed through a Multi-Layer Perceptron (MLP) to extract relevant information and consider $t$-dimensional readouts for these topological encodings.

Finally, we combine the GNN-updated positional embeddings with the MLP-processed topological embeddings using another MLP, updating the combined embeddings in an end-to-end fashion. This approach leverages both global and local graph information, aiming to improve the expressiveness and accuracy of the GNN model. The flowchart of our framework is shown in Figure 1.

## 4 EXPERIMENTS

In this section, we assess the performance of our framework in the node classification setting. It's important to note that our framework offers significant flexibility, allowing for the adoption of any GNN model based on message passing with our WISE embeddings.

### 4.1 EXPERIMENTAL SETUP

We show the effectiveness of our framework in node classification and visualization tasks.

**Datasets.** We conducted experiments on three homophilic graph datasets of CORA, CITESEER, and PUBMED, and four heterophilic datasets: TEXAS, CORNELL, WISCONSIN, and CHAMELEON, as proposed by (PWC$^+$19). Detailed descriptions, statistics, and homophily measures for these datasets can be found in Table 1.

Table 1: Benchmark datasets for node classification.

| Datasets | Nodes | Edges | Class | Features | Hom. |
|---|---|---|---|---|---|
| CORA | 2,708 | 5,429 | 7 | 1,433 | 0.83 |
| CITESEER | 3,312 | 4,732 | 6 | 3,703 | 0.72 |
| PUBMED | 19,717 | 44,338 | 3 | 500 | 0.79 |
| TEXAS | 183 | 309 | 5 | 1,703 | 0.10 |
| CORNELL | 183 | 295 | 5 | 1,703 | 0.39 |
| WISCONSIN | 251 | 499 | 5 | 1,703 | 0.15 |
| CHAMELEON | 2,277 | 36,101 | 5 | 2,325 | 0.25 |

**Models.** We utilized our WISE-GNN with three classical models, GCN (KW17), GraphSAGE (HYL17), and GAT (VCC$^+$18). In addition, we included LINKX (LHL$^+$21) and H2GCN (ZYZ$^+$20) in our evaluation. Notably, LINKX is specifically designed for handling heterophilic datasets. In Table 2, we provide performances of three different variations (TGNN, WGNN and TWGNN) as well as vanilla model GNN. We use a 3-layer MLP model on node feature vectors as a baseline. TGNN represents vanilla GNN + Topological Module. In WGNN, we replace the initial node embeddings (node feature vectors) with our Wise embeddings. TWGNN means WGNN is incorporated with the Topological Module.

**Parameters.** To optimize the performance of both the GNN and the first MLP embeddings, we conduct hyperparameter tuning. Through experimentation with different dimensional embeddings, we find that the WISE-GNN embeddings perform optimally when GNN output dimensionality is greater than the number of classes and the MLP output embeddings dimensionality is less than the number of classes. So we set he dimension as $w = 10$ for the GNN embeddings and $t = 5$ for the MLP embeddings throughout the experiment.

This choice is informed by the understanding that while positional encodings are powerful on their own, topological encodings may be less so, as mentioned in Table 4. Therefore, we opt for a lower number of dimensions for the topological encodings to achieve better performance. Through this tuning process, we aim to find the optimal balance between expressive power and computational efficiency for both types of embeddings, ultimately enhancing the overall performance of our WISE-GNN framework.

**Implementation.** To ensure a fair comparison, we maintain the same experimental setup for both the GNN and WISE-GNN models. For our WISE-GNN, we incorporate an MLP layer to provide topological or structural information to the GNN model, enabling end-to-end training. This setup allows our framework to leverage both the expressive power of GNNs and the additional insights provided by the MLP layer. We implement a two-layer GNN framework, following the methodology

Table 2: **GNN Improvements.** For each GNN backbone, we present the node classification accuracy results for three variants: *T-GNN* (with a topological module), *W-GNN* (initialized with Wise embeddings), and *TW-GNN* (combining both). **Av. ↑** indicates the average improvement over the vanilla model for homophilic and heterophilic datasets. The best performance for each GNN model is highlighted in **bold**.

| GNN | Model | CORA 0.83 | CITESEER 0.72 | PUBMED 0.79 | Av. ↑ | TEXAS 0.10 | CORNELL 0.39 | WISC 0.15 | CHAM 0.25 | Av. ↑ |
|---|---|---|---|---|---|---|---|---|---|---|
| GCN | GCN | $74.77_{\pm2.41}$ | $59.03_{\pm3.01}$ | $73.95_{\pm1.86}$ | – | $56.49_{\pm8.87}$ | $45.68_{\pm6.55}$ | $53.14_{\pm4.18}$ | $65.88_{\pm1.79}$ | – |
| | T-GCN | $75.31_{\pm1.75}$ | $62.43_{\pm2.08}$ | $75.93_{\pm0.90}$ | 1.97 | $58.11_{\pm7.78}$ | $46.65_{\pm5.17}$ | $56.08_{\pm7.46}$ | **$66.93_{\pm1.91}$** | 1.65 |
| | W-GCN | $84.35_{\pm1.17}$ | $77.20_{\pm0.91}$ | $75.33_{\pm2.51}$ | 9.71 | $72.97_{\pm8.92}$ | $60.81_{\pm9.89}$ | $60.78_{\pm6.27}$ | $61.32_{\pm3.22}$ | 7.19 |
| | TW-GCN | **$85.71_{\pm0.68}$** | **$78.19_{\pm1.82}$** | **$76.36_{\pm2.47}$** | 10.84 | **$73.51_{\pm5.22}$** | **$61.35_{\pm7.76}$** | **$62.35_{\pm3.90}$** | $63.03_{\pm3.15}$ | 9.76 |
| GSAGE | GSAGE | $69.34_{\pm3.02}$ | $50.39_{\pm3.04}$ | $70.93_{\pm1.49}$ | – | $75.41_{\pm7.26}$ | $61.89_{\pm5.47}$ | $71.57_{\pm5.25}$ | $62.61_{\pm2.75}$ | – |
| | T-GSAGE | $75.25_{\pm2.84}$ | $55.42_{\pm5.39}$ | $73.37_{\pm1.01}$ | 4.46 | $75.78_{\pm7.54}$ | $61.62_{\pm5.95}$ | $70.20_{\pm4.22}$ | $65.50_{\pm1.39}$ | 0.41 |
| | W-GSAGE | $83.50_{\pm1.20}$ | $75.37_{\pm1.96}$ | $73.22_{\pm3.12}$ | 13.81 | $88.38_{\pm4.42}$ | $81.89_{\pm5.85}$ | $87.25_{\pm5.49}$ | $76.32_{\pm1.96}$ | 15.60 |
| | TW-GSAGE | **$84.40_{\pm0.97}$** | **$76.94_{\pm1.19}$** | **$73.99_{\pm2.63}$** | 14.89 | **$92.97_{\pm2.61}$** | **$84.32_{\pm5.52}$** | **$90.98_{\pm4.15}$** | **$78.90_{\pm1.29}$** | 18.92 |
| GAT | GAT | $74.71_{\pm1.49}$ | $60.24_{\pm2.00}$ | $74.01_{\pm1.69}$ | – | $55.14_{\pm6.52}$ | $43.78_{\pm7.41}$ | $52.75_{\pm7.87}$ | $64.82_{\pm2.15}$ | – |
| | T-GAT | $76.16_{\pm1.84}$ | $62.32_{\pm1.58}$ | $75.76_{\pm0.65}$ | 1.76 | $57.57_{\pm7.10}$ | $44.32_{\pm5.73}$ | $53.53_{\pm6.41}$ | **$66.73_{\pm2.03}$** | 1.42 |
| | W-GAT | $83.51_{\pm0.92}$ | $76.71_{\pm1.74}$ | $75.34_{\pm2.93}$ | 8.87 | $62.97_{\pm8.83}$ | $47.57_{\pm5.87}$ | $54.31_{\pm8.11}$ | $54.25_{\pm7.70}$ | 0.65 |
| | TW-GAT | **$84.93_{\pm1.07}$** | **$77.53_{\pm1.38}$** | **$76.56_{\pm2.48}$** | 10.02 | **$66.22_{\pm6.14}$** | **$53.51_{\pm6.60}$** | **$60.00_{\pm6.74}$** | $56.77_{\pm5.44}$ | 5.00 |
| LINKX | LINKX | $49.46_{\pm3.82}$ | $42.94_{\pm1.23}$ | $66.52_{\pm1.21}$ | – | $73.24_{\pm6.30}$ | $72.43_{\pm7.18}$ | $80.00_{\pm6.97}$ | $63.57_{\pm2.93}$ | – |
| | T-LINKX | $59.52_{\pm2.27}$ | $54.93_{\pm2.28}$ | $69.50_{\pm1.50}$ | 8.34 | $80.81_{\pm4.50}$ | $75.24_{\pm4.50}$ | $85.69_{\pm3.93}$ | $64.76_{\pm1.87}$ | 4.32 |
| | W-LINKX | $63.42_{\pm2.88}$ | $68.20_{\pm2.73}$ | $67.34_{\pm2.92}$ | 13.34 | $90.95_{\pm2.63}$ | $89.70_{\pm4.24}$ | $90.35_{\pm5.18}$ | $83.76_{\pm2.33}$ | 16.38 |
| | TW-LINKX | **$64.81_{\pm3.41}$** | **$69.04_{\pm1.89}$** | **$69.70_{\pm2.35}$** | 14.87 | **$92.97_{\pm2.91}$** | **$90.27_{\pm4.80}$** | **$91.57_{\pm3.46}$** | **$84.36_{\pm1.70}$** | 17.48 |
| H2GCN | H2GCN | $77.76_{\pm1.55}$ | $62.83_{\pm2.01}$ | $74.26_{\pm2.12}$ | – | $72.43_{\pm4.73}$ | $66.22_{\pm6.53}$ | $75.88_{\pm4.81}$ | $49.89_{\pm2.49}$ | – |
| | T-H2GCN | $69.39_{\pm4.04}$ | $55.27_{\pm2.57}$ | $71.30_{\pm1.32}$ | -6.29 | $74.32_{\pm4.12}$ | $69.65_{\pm5.58}$ | $82.94_{\pm5.07}$ | $52.51_{\pm2.87}$ | 3.75 |
| | W-H2GCN | **$82.26_{\pm1.49}$** | $77.16_{\pm2.23}$ | **$76.50_{\pm1.42}$** | 7.02 | $85.14_{\pm10.20}$ | $82.43_{\pm7.01}$ | $84.90_{\pm7.34}$ | $77.96_{\pm2.02}$ | 16.50 |
| | TW-H2GCN | $80.57_{\pm1.30}$ | **$78.68_{\pm2.00}$** | $75.43_{\pm1.20}$ | 6.61 | **$92.97_{\pm3.86}$** | **$92.43_{\pm3.07}$** | **$92.75_{\pm3.82}$** | **$80.83_{\pm3.86}$** | 23.64 |

outlined in (KW17), utilizing Adam optimization with a learning rate of $0.01$. The hyper-parameter settings include a dropout rate of $p = 0.5$, an initial learning rate of $0.01$, and weight decay of $5E - 6$. The number of hidden channels is set to 32 for PUBMED, and 16 for the remaining datasets, chosen to balance model complexity and performance across diverse datasets. For the MLP framework, we employ a three-layer MLP with 100 hidden channels, allowing for the extraction of higher-level features from the input data. Dropout regularization is applied, with dropout rates chosen from 0, 0.5, to prevent overfitting and improve generalization. We optimize the MLP using the Adam optimizer with a learning rate of 0.01, facilitating efficient training and convergence. All models are trained for a maximum of 200 epochs (training iterations), ensuring sufficient exploration of the parameter space and convergence to stable solutions. This extended training period enables the models to capture intricate patterns and relationships present in the input data, leading to improved performance on various tasks and datasets.

For homophilic graphs, we utilize publicly available splits consisting of 20 nodes per class for training, 500 nodes for validation, and 1,000 nodes for testing (KW17). For heterophilic graphs, we adhere to the commonly used training/validation/test split ratio of $48/32/20$, consistent with previous works (PWC⁺19). We conducted our experiments using Python, and the code is available at `https://anonymous.4open.science/r/Topo_Wise_GNN-EE32`.

## 4.2 NODE CLASSIFICATION RESULTS

We present our results across two tables (Tables 2 and 3). First, Table 2 illustrates the significant impact of Wise embeddings and the topological module on the performance of five different GNN architectures. In particular, Wise embeddings consistently enhance the performance of all five GNN models, with average accuracy gains ranging from 7% to 16% (with the exception of GAT on heterophilic datasets). While the topological module alone provides moderate improvements, combining it with Wise embeddings leads to consistent performance boosts across all models, achieving an additional 1% to 7% increase in accuracy (except for H2GCN on homophilic datasets).

At the individual dataset level, Wise-GNN demonstrates significant accuracy gains on heterophilic datasets, with improvements ranging from 10% to 30%. The only exceptions occur with the Chameleon dataset when using GCN and GAT, though even in these cases, incorporating topological information still enhances performance. This underscores the value of leveraging graph topology for more effective representation learning. On homophilic datasets, Wise-GNN also shows notable improvements: 10% to 15% on CORA (except for H2GCN), 15% to 25% on CITESEER, and 2% to 3% on PUBMED. The limited improvement for H2GCN on homophilic datasets is likely due

Table 3: **SOTA baselines.** Node classification accuracy results of SOTA baselines and our TW-GNN models, utilizing different backbones. The top-performing baselines are highlighted in blue, while the best accuracy for each dataset is shown in **bold**. **Red columns** shows the average deviation of each model's performance from the best performance across (1) homophilic, (2) heterophilic, and (3) all datasets.

| Model | CORA 0.83 | CITESEER 0.72 | PUBMED 0.79 | Hom Av. ↓ | TEXAS 0.10 | CORNELL 0.39 | WISC 0.15 | CHAM 0.25 | Het Av. ↓ | All Av. ↓ |
|---|---|---|---|---|---|---|---|---|---|---|
| GCA (ZXY$^+$21) | $82.93_{\pm0.42}$ | $72.19_{\pm0.31}$ | $80.79_{\pm0.45}$ | 3.8 | $52.92_{\pm0.46}$ | $52.31_{\pm1.09}$ | $59.55_{\pm0.81}$ | $63.66_{\pm0.32}$ | 33.5 | 20.8 |
| CCA-SSG (ZWY$^+$21) | $84.00_{\pm0.40}$ | $73.10_{\pm0.30}$ | $81.00_{\pm0.40}$ | 3.1 | $59.89_{\pm0.78}$ | $52.17_{\pm1.04}$ | $58.46_{\pm0.96}$ | $62.41_{\pm0.22}$ | 32.4 | 19.8 |
| BGRL (TTA$^+$21) | $82.70_{\pm0.60}$ | $71.10_{\pm0.80}$ | $79.60_{\pm0.50}$ | 4.6 | $52.77_{\pm1.98}$ | $50.33_{\pm2.29}$ | $51.23_{\pm1.17}$ | $64.86_{\pm0.63}$ | 35.8 | 22.5 |
| L-GCL (ZWW$^+$22) | $84.00_{\pm0.35}$ | $73.26_{\pm0.50}$ | $81.82_{\pm0.50}$ | 2.7 | $60.68_{\pm1.18}$ | $52.11_{\pm2.37}$ | $65.28_{\pm0.52}$ | $68.74_{\pm0.49}$ | 28.9 | 17.7 |
| HGRL (CZQ$^+$22) | $82.52_{\pm0.31}$ | $71.05_{\pm0.49}$ | $79.83_{\pm0.31}$ | 4.6 | $61.83_{\pm0.71}$ | $51.78_{\pm1.03}$ | $63.90_{\pm0.58}$ | $65.82_{\pm0.61}$ | 29.8 | 19.0 |
| DSSL (XCG$^+$22) | $83.51_{\pm0.42}$ | $73.20_{\pm0.51}$ | $81.25_{\pm0.31}$ | 3.1 | $62.11_{\pm1.53}$ | $53.15_{\pm1.28}$ | $62.25_{\pm0.55}$ | $66.15_{\pm0.32}$ | 29.7 | 18.3 |
| GREET (LZZ$^+$23) | $83.81_{\pm0.87}$ | $73.08_{\pm0.84}$ | $80.29_{\pm1.00}$ | 3.4 | $87.00_{\pm\text{ NA}}$ | $85.10_{\pm\text{ NA}}$ | $84.90_{\pm\text{ NA}}$ | $63.60_{\pm\text{ NA}}$ | 10.5 | 7.4 |
| MUSE (YCL23) | $82.24_{\pm0.24}$ | $71.14_{\pm0.40}$ | $\mathbf{82.90_{\pm0.59}}$ | 3.6 | $89.73_{\pm2.79}$ | $82.16_{\pm3.42}$ | $88.24_{\pm3.20}$ | $72.37_{\pm2.21}$ | 7.5 | 5.9 |
| SP-GCL (WZZ$^+$24) | $83.16_{\pm0.13}$ | $71.96_{\pm0.42}$ | $79.16_{\pm0.73}$ | 4.3 | $59.81_{\pm1.33}$ | $52.29_{\pm1.21}$ | $60.12_{\pm0.39}$ | $65.28_{\pm0.53}$ | 31.3 | 19.7 |
| DHGR (BDF$^+$24) | $82.12_{\pm0.49}$ | $70.87_{\pm0.29}$ | $79.65_{\pm0.58}$ | 4.9 | $84.86_{\pm5.01}$ | $82.06_{\pm6.27}$ | $85.01_{\pm5.51}$ | $69.19_{\pm1.93}$ | 10.3 | 8.0 |
| GraphACL (XZCW24) | $84.20_{\pm0.31}$ | $73.63_{\pm0.22}$ | $82.02_{\pm0.15}$ | 2.5 | $71.08_{\pm0.34}$ | $59.33_{\pm1.48}$ | $69.22_{\pm0.40}$ | $69.12_{\pm0.24}$ | 23.4 | 14.5 |
| TEDGCN (YCC$^+$24) | $82.50_{\pm1.10}$ | $70.80_{\pm0.70}$ | $79.20_{\pm0.20}$ | 4.9 | $77.60_{\pm5.90}$ | $72.00_{\pm5.80}$ | $82.00_{\pm2.60}$ | $55.70_{\pm1.30}$ | 18.8 | 12.9 |
| TW-GCN | $\mathbf{85.71_{\pm0.68}}$ | $78.19_{\pm1.82}$ | $76.36_{\pm2.47}$ | 2.3 | $73.51_{\pm5.22}$ | $61.35_{\pm7.76}$ | $62.35_{\pm3.90}$ | $63.03_{\pm3.15}$ | 25.6 | 15.6 |
| TW-GSAGE | $84.40_{\pm0.97}$ | $76.94_{\pm1.19}$ | $73.99_{\pm2.63}$ | 4.0 | $92.97_{\pm2.61}$ | $84.32_{\pm5.52}$ | $90.98_{\pm4.15}$ | $78.90_{\pm1.29}$ | 3.8 | 3.9 |
| TW-GAT | $84.93_{\pm1.07}$ | $77.53_{\pm1.38}$ | $76.56_{\pm2.48}$ | 2.8 | $66.22_{\pm6.14}$ | $53.51_{\pm6.60}$ | $60.00_{\pm6.74}$ | $56.77_{\pm5.44}$ | 31.5 | 19.2 |
| TW-LINKX | $64.81_{\pm3.41}$ | $69.04_{\pm1.89}$ | $69.70_{\pm2.35}$ | 14.6 | $92.97_{\pm2.91}$ | $90.27_{\pm4.80}$ | $91.57_{\pm3.46}$ | $\mathbf{84.36_{\pm1.70}}$ | 0.8 | 6.7 |
| TW-H2GCN | $80.57_{\pm1.30}$ | $\mathbf{78.68_{\pm2.00}}$ | $75.43_{\pm1.20}$ | 4.9 | $\mathbf{92.97_{\pm3.86}}$ | $\mathbf{92.43_{\pm3.07}}$ | $\mathbf{92.75_{\pm3.82}}$ | $80.83_{\pm3.86}$ | 0.9 | 2.6 |

to its partial integration of GCN's backbone, reducing the impact of further enhancements. However, the substantial gains on heterophilic datasets demonstrate the strength of our distant node learning strategies. Finally, it's important to note that while GCN, GraphSAGE, and GAT are predominantly designed for homophilic datasets, Wise-GNN—thanks to its Wise embeddings and topological encoding—still delivers performance improvements in homophilic contexts.

Second, in Table 3, we compare the SOTA baselines with our WISE-GNN models (TW-GNN). Across all experiments, WISE-GNN models show outstanding performance. For homophilic datasets, WISE-GNN models surpass the SOTA baselines in all cases except for the PUBMED dataset and the TW-LINKX model. In heterophilic datasets, WISE-GNN also outperforms several recent GNN models specifically designed for heterophilic data. Across heterophilic datasets, WISE-GNN improves SOTA performances by 3% to 12%, representing a significant advancement in the field.

While GSAGE is considered a traditional GNN model, we observe that TW-GSAGE outperforms all SOTA results on every dataset except PUBMED. This highlights the effectiveness of the WISE-GNN models. Given the ease of integrating both Wise-embeddings and the topological module into any GNN model, we believe our approach can be effectively applied to future GNN models to address the inherent limitations previously discussed.

In conclusion, WISE-GNN often outperforms or is at least comparable to state-of-the-art methods, while offering broad applicability to a range of GNN architectures. Its ability to efficiently handle both homophilic and heterophilic datasets makes it a versatile and powerful solution for a wide variety of graph learning tasks.

## 4.3 VISUALIZING NODE REPRESENTATIONS

In this section, we demonstrate how WISE-GNN embeddings enhance GNNs in generating superior node representations, as illustrated by t-SNE (VdMH08) visualization plots. Figure 2 show node representation for the CORA and WISCONSIN datasets where data points (nodes) are colored according to their classes. Figures 2a to 2d demonstrates the initial and final embeddings for the GCN and W-GCN models, where we observed that the classes are not well separated in the GCN-initial embedding. While in the Wise embedding (W-GCN initial), the classes are much better separated, demonstrating that our model leads to a more refined and meaningful node representation in the final W-GCN visualization. Next, Figure 2 shows similar results for the GraphSage model; W-GSAGE final embeddings have better separation of classes in the center. These results underscore the importance of measuring the proximity of nodes in the latent space.

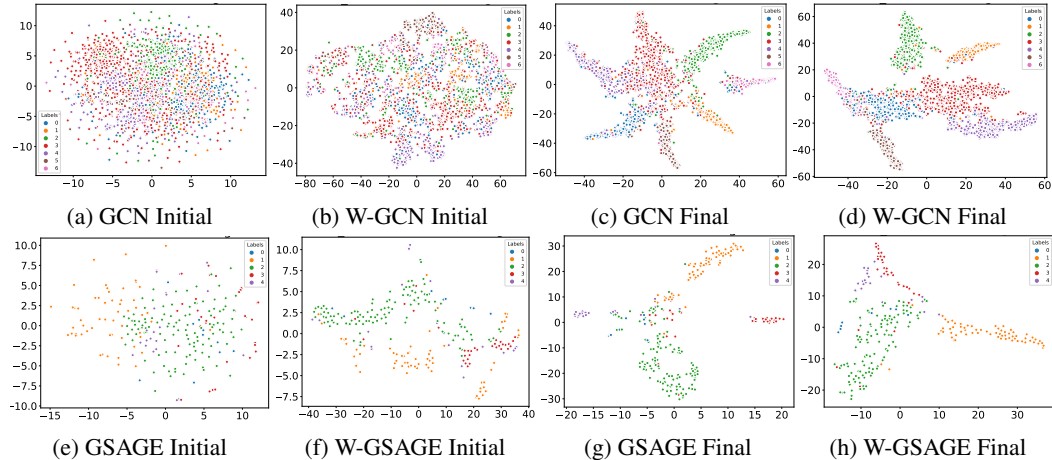

(a) GCN Initial     (b) W-GCN Initial     (c) GCN Final     (d) W-GCN Final

(e) GSAGE Initial     (f) W-GSAGE Initial     (g) GSAGE Final     (h) W-GSAGE Final

Figure 2: **Visualization of Node Embeddings.** t-SNE visualization of the CORA (a-d) and WISCONSIN (e-h) datasets. The final embeddings represent the best embeddings from the GSAGE and W-GSAGE models, corresponding to the highest validation accuracy achieved.

## 4.4 ABLATION STUDIES

To verify the effectiveness of our method, we conducted three ablation studies. In the first study, we evaluated the performance of GCN with different initial embeddings. In the second study, we compared the performance of various similarity measures used to define classWISE landmarks. Finally, in the third study, we examined the impact of the size and removing the center of the node neighborhoods on the performance of topological embeddings.

**Initial Embeddings.** In our model, we used Wise Embeddings as the initial vectors and integrated topological encodings through an MLP. A natural question arises: why not use both as initial embeddings? We aim to answer this question through our ablation study. As shown in Table 4, we examined the performance of the GCN model with different initial embeddings. The first row shows the original GCN model using node feature vectors as initial embeddings. In the second row, we see a significant performance improvement when replacing node feature vectors with Wise Embeddings. However, in the third row, when using only topological encodings as initial vectors, the performance drops below that of the original GCN model (NF).

The most surprising and crucial observation is in the fourth row. When we concatenate Wise Embeddings with topological encodings as initial embeddings, there is a significant performance drop. Intuitively, adding topological encodings to Wise Embeddings should enhance performance. However, we found that Wise Embeddings contain the key information for the GCN, and while topological encodings add extra information, they also dilute the crucial information from Wise Embeddings due to the additional dimensions. This is why we chose to use Wise Embeddings as the initial embeddings and integrate topological encodings through the MLP. This approach ensures that Wise Embeddings remain intact while topological encodings still contribute. Comparing the fourth row of Table 4 with the fourth row of Table 2 shows a significant performance difference. Similarly, the last row indicates that while original node features add some value, they also harm performance due to the same issue.

Table 4: **The Effect of Initial Embeddings.** Node classification accuracy results of GCN models using different vectors as initial embeddings. Below, NF is node feature vector, WE is Wise Embeddings, and TE is Topological Encodings. Multiple options mean the concatenation of the corresponding vectors.

| NF | WE | TE | CORA | CITESEER | PUBMED | TEXAS | CORNELL | WISCONSIN |
|----|----|----|------|----------|--------|-------|---------|-----------|
| ✓ | ✗ | ✗ | $74.77_{\pm 2.41}$ | $59.03_{\pm 3.01}$ | $73.95_{\pm 1.86}$ | $56.49_{\pm 8.87}$ | $45.68_{\pm 6.55}$ | $53.14_{\pm 4.18}$ |
| ✗ | ✓ | ✗ | $\mathbf{84.35_{\pm 1.17}}$ | $\mathbf{77.20_{\pm 0.91}}$ | $\mathbf{75.33_{\pm 2.51}}$ | $\mathbf{72.97_{\pm 8.92}}$ | $\mathbf{60.81_{\pm 9.89}}$ | $\mathbf{60.78_{\pm 6.27}}$ |
| ✗ | ✗ | ✓ | $44.05_{\pm 0.78}$ | $38.22_{\pm 1.42}$ | $46.36_{\pm 5.47}$ | $57.84_{\pm 6.27}$ | $45.41_{\pm 4.19}$ | $51.96_{\pm 4.36}$ |
| ✗ | ✓ | ✓ | $73.82_{\pm 5.31}$ | $75.63_{\pm 0.56}$ | $56.45_{\pm 1.23}$ | $62.70_{\pm 8.53}$ | $51.08_{\pm 5.90}$ | $58.82_{\pm 4.53}$ |
| ✓ | ✓ | ✓ | $78.17_{\pm 1.96}$ | $65.63_{\pm 1.44}$ | $62.49_{\pm 1.54}$ | $61.35_{\pm 6.63}$ | $45.68_{\pm 6.04}$ | $58.24_{\pm 5.55}$ |

**Similarity Measures.** In Wise Embeddings, the landmark identification and similarity measures used are crucial for the effectiveness of the vectors. When the attribute vectors are all binary, Jaccard similarity, Positive similarity, and Cosine similarity are among the most common methods to measure the distance similarity of node embeddings to landmarks for Wise embeddings (see Appendix A). As shown in Table 5, Positive similarity showed the best performance, which we employed in our model.

Table 5: Classification accuracy of W-GCN with different similarity measures on CORA and TEXAS datasets.

| Similarity Measure | CORA | TEXAS |
|---|---|---|
| Jaccard Similarity | $77.09\pm1.47$ | $67.30\pm7.37$ |
| Positive Similarity | **$84.35\pm1.17$** | **$72.97\pm8.92$** |
| Cosine Similarity | $81.94\pm0.91$ | $68.38\pm6.63$ |

**Neighborhood Sizes and Structures.** In our second ablation study, we applied the W-GCN model to analyze the importance and utility of different hop neighborhoods on CORA and TEXAS datasets, as outlined in Table 6. When considering both effectiveness and time complexity, we found that the 2-hop neighborhood yielded optimal results for our model. For topological signatures, we tested the impact of neighborhood size and the effect of removing the central node. From the perspective of persistent homology, removing the central node from the neighborhood can significantly increase the number of topological features. The size of the neighborhood also plays a crucial role. We evaluated these factors in our topological signatures and reported the results in Table 6. While the performances are generally similar, we observed a significant change in the no-ego 1-hop and 2-hop neighborhoods.

Overall, the empirical evaluation highlights the effectiveness of our WISE-GNN model in harnessing both the positional and topological information to enhance performance in node classification tasks.

Table 6: Performance Evaluation and Computational complexity of WISE-GNN Using Topological Features Across Different Hop Neighborhoods

| Dataset | Ego/Center | 1-hop | Time (s) | 2-hop | Time (s) | 3-hop | Time (s) |
|---|---|---|---|---|---|---|---|
| CORA | Yes | $84.51\pm2.04$ | 66.69 | $84.93\pm1.99$ | 70.29 | $84.79\pm0.65$ | 81.58 |
| | No | $84.56\pm1.19$ | 64.43 | $84.73\pm1.32$ | 71.84 | $84.63\pm1.25$ | 82.99 |
| TEXAS | Yes | $72.97\pm6.37$ | 10.00 | $73.51\pm5.22$ | 10.22 | $72.43\pm8.62$ | 10.72 |
| | No | $72.70\pm5.47$ | 9.98 | $75.41\pm5.76$ | 10.23 | $72.70\pm6.68$ | 10.75 |

**Limitations.** While our model effectively incorporates positional encodings of nodes in feature space and furnishes GNNs with crucial information about node representations, the primary challenge lies in determining the optimal landmarks and similarity measures to represent class identifiers. In this paper, we proposed various common similarity methods and landmark selection techniques, and combinations of these methods performed well across all datasets. In future work, we plan to make these processes learnable for the downstream tasks and create an end-to-end GNN framework.

## 5 CONCLUSION

In this paper, we tackle the challenges identified in recent theoretical studies regarding the limitations of GNNs, particularly issues such as oversquashing and underreaching. By integrating information from relevant but distant nodes during the initial embedding stage, we demonstrate that Wise Embeddings significantly improve the performance of various GNN models by incorporating essential insights from the node attribute space into the message-passing framework.

Building on these findings, our future work will develop models that can extend the embeddings to the temporal graph learning domain, where evolving distances can create graph embeddings to understand complex system behavior.

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
