APPENDIX

## A  SIMILARITY MEASURES

When comparing two binary arrays representing node features with $n$ binary attributes, we compute four quantities, often referred to as frequencies, from the given binary data:

- $a$ – The number of attributes that are equal to 1 for both objects i and j.
- $b$ – The number of attributes that are equal to 0 for object i but equal to 1 for object j.
- $c$ – The number of attributes that are equal to 1 for object i but equal to 0 for object j.

Based on these frequencies, several popular similarity measures are defined as follows:

- **Jaccard Similarity** is defined as $\mathcal{J}(i,j) = \dfrac{a}{a+b+c}$
- **Positive Similarity** is defined as $\mathcal{P}(i,j) = a$
- **Cosine Similarity** is defined as $\mathcal{C}(i,j) = \dfrac{a}{\sqrt{(a+b)(a+c)}}$

Table 5 displays the performance of the W-GCN model across various similarity measures. Following an assessment of these measures, we have chosen *positive similarity* for this study, particularly suited for datasets containing binary node feature vectors.

## B  CLASSWISE LANDMARKS FOR EACH DATASET

Our landmark selection methods are highly flexible and can be applied to various datasets with different attribute vector formats. In this section, we provide details on how these methods were implemented in the benchmark datasets.

**CORA:** The Cora dataset is a widely used benchmark in the field of machine learning, particularly for tasks involving graph neural networks and semi-supervised learning. It consists of a citation network of scientific publications classified into one of seven classes. The dataset contains 2,708 nodes, each representing a document, and 5,429 edges, each representing a citation link between two documents. Each document is described by a 1,433-dimensional binary word vector, indicating the absence or presence of specific words from a predefined dictionary. Since the dataset has categorical domain features, we consider two class-wise landmarks: one is inclusive and the other is selective. The inclusive landmark, $\xi_i^j$, is a binary word vector of length 1433, where each entry is 1 if the corresponding word is present in any binary vectors $X_u$ of class $C_j$, and 0 otherwise. For the selective landmark vector, $\xi_s^j$ is defined as a binary word vector indicating the presence of the corresponding word in at least 10% of nodes in class $C_j$. Thus, we have 7 class-wise landmarks for the inclusive choice and another 7 class-wise landmarks for the selective choice, totaling 14 class-wise landmarks. By considering the similarity measure with these landmarks for a given node, we can create a 14-dimensional Positional embedding for the TWGNN model.

**CITESEER:** It consists of a citation network of scientific publications classified into one of six classes. The dataset contains 3,327 nodes, each representing a document, and 4,732 edges, each representing a citation link between two documents. Each document is described by a 3,703-dimensional binary word vector, indicating the absence or presence of specific words from a predefined dictionary. Since the dataset shares similar characteristics with the Cora dataset, we employ the same technique for classWISE landmarks selection. This involves creating six inclusive and six selective landmarks, resulting in a total of twelve landmarks. These landmarks capture important information about the presence of specific words within each class of nodes. Subsequently, we utilize these landmarks to generate a twelve-dimensional positional embedding for our model.

**PUBMED:** The PUBMED dataset consists of a directed graph containing 19,717 scientific publications obtained from the PubMed database, specifically targeting diabetes research, and divided into three classes. Each node within the graph represents a publication and is defined by a TF/IDF weighted word vector derived from a vocabulary of 500 unique words. As the domain features are

real-valued, we select three landmarks by computing the mean value along each dimension of the word vectors, resulting in three 500-dimensional landmarks. Utilizing the Euclidean distance as a similarity measure, we construct a three-dimensional initial positional embedding for our model, enabling the effective representation of the dataset's structural and textual characteristics.

**TEXAS, CORNELL, and WISCONSIN:** The Texas, Cornell, and Wisconsin datasets all represent web-page networks, where nodes correspond to web pages and edges signify hyperlinks between them. The Wisconsin dataset comprises 251 nodes, while both the Cornell and Wisconsin datasets contain 183 nodes each. Each node is characterized by a feature vector of 1,703 binary attributes, denoting the presence or absence of specific words within the web page content. Additionally, the nodes are classified into one of five categories based on the type of web page they represent. Given that the feature vector is categorical, akin to the CORA dataset, we can employ a similar technique for feature extraction. This involves creating five selective and five inclusive landmarks. Consequently, we can generate a ten-dimensional positional embedding for our model.

**CHAMELEON:** The datasets represent page-page networks centered around specific topics such as chameleons. In these networks, nodes correspond to articles, while edges signify mutual links between them. Node features are extracted from informative nouns found in the corresponding Wikipedia pages. The presence of a feature in the feature list indicates the occurrence of an informative noun in the text of the Wikipedia article. These two datasets also share categorical features and possess feature vectors of dimensions 2325 and 2089 respectively. Both datasets are structured into five classes. Employing our categorical class-wise landmark selection technique, we establish five inclusive and five selective landmarks with dimensions 2325 and 2089.