# OpenReview forum: "WISE-GNN: Enhancing GNNs with Wise Embedding and Topological Encoding"
_ICLR.cc/2025/Conference — ICLR 2025 Conference Withdrawn Submission_

### Official Review · Reviewer_7h2H · 2024-11-02

**Soundness:** 2
**Presentation:** 1
**Contribution:** 2
**Rating:** 3
**Confidence:** 4

**Summary:**

This paper introduces a framework called **WISE-GNN**, designed to enhance the performance of Graph Neural Networks (GNNs) in node classification tasks, particularly in **heterophilic graphs**. WISE-GNN improves node representation by incorporating "Wise embeddings" and a topology encoding module. These enhancements address the challenges faced by GNNs in capturing long-range dependencies between distant nodes.

**Strengths:**

- The design of WISE-GNN is model-agnostic, allowing it to be easily integrated with any existing GNN architecture.

**Weaknesses:**

### **Method**
1. The wise-gnn obtains new features by measuring the distance between nodes and multiple cluster centers, making the representations of similar initial nodes more alike. Node representations are updated through a message-passing mechanism. If a node has a high degree of heterophilic, the message-passing approach will still influence the final node representation.

2. The introduction of the topology module does not effectively interact with the wise embedding, but rather functions as a simple combination.

### **Experiment**

1. The dataset used in the paper is too small.
2. The method lacks comparision with the strong baselines for heterophilic graphs.
3. The performance of GCN on Cora is too low on the public split.

**Questions:**

- Discuss the advantages of WISE-GNN compared to Geom-GCN[1], which utilizes a method for obtaining structural neighbors from a distance, as Geom-GCN can overcome the limitations of message-passing.
- Is there a better way to combine the representations in the topology module?
- How does WISE-GNN perform on large graphs, such as Ogbn-Arxiv and Ogbn-Products?
- As pointed out in paper[2], the heterophilic graph (Chameleon)  has an information leakage issue. How does WISE-GNN perform on currently widely used (larger and diverse properties) heterophilic graphs[2]?
- Can you compare it with the strong heterophilic graph baselines, such as ACM-GCN [3], GREAD[4], M2M-GNN[5] etc.?
- I would appreciate it if you could explain the results of GCN on Cora?


[1] Geom-GCN: Geometric Graph Convolutional Networks, ICLR 2020

[2] A critical look at the evaluation of GNNs under heterophily: Are we really making progress? NeurIPS 2023

[3] Revisiting Heterophily For Graph Neural Networks, NeurIPS 2022

[4]GREAD: Graph Neural Reaction-Diffusion Networks, ICML 2023

[5] Sign is Not a Remedy: Multiset-to-Multiset Message Passing for Learning on Heterophilic Graphs, ICML 2024

---

### Official Review · Reviewer_oVsh · 2024-11-03

**Soundness:** 2
**Presentation:** 2
**Contribution:** 2
**Rating:** 5
**Confidence:** 4

**Summary:**

The paper introduces WISE-GNN, a framework aimed at improving GNN performance in capturing long-range dependencies on graphs, particularly in heterophilic settings. This is achieved through two main components: Wise embeddings, which capture node attribute similarities to class-specific landmarks, and a Topological Module based on persistent homology to encode local subgraph structures.

**Strengths:**

1. WISE-GNN addresses both attribute and topological aspects of graph data, aiming for a more comprehensive representation that could potentially aid tasks in complex graph structures.
2. The framework is model-agnostic, making it adaptable to various GNN architectures with minimal modifications.

**Weaknesses:**

1. Unclear Objective Function: The objective function and the overall process for integrating Wise embeddings into the GNN model are insufficiently explained, which hinders understanding of how these embeddings work in the training framework.
2. Limited Motivation for Wise Embeddings: While capturing information from distant nodes is valuable, the use of classWISE benchmarks as an effective means of encoding this information lacks sufficient justification. It’s unclear why this specific method would be more beneficial than alternative strategies, such as clustering approaches, which may offer richer structural context.
3. Restricted Novelty: The Topological Module is primarily adopted from established work in persistent homology, offering limited originality in its formulation or integration. The Wise embedding concept also appears somewhat incremental rather than ground-breaking.
4. Experimental Ambiguity: The experimental setup and results need more clarity, particularly concerning the T-H2GCN performance and whether the results fully validate the claimed improvements. Additionally, comparisons with clustering-based methods would strengthen the case for Wise embeddings as a distinct improvement.

**Questions:**

1. Could the authors clarify the objective function and explain how Wise embeddings interact with GNN updates during training?
2. Why were Wise embeddings, rather than more conventional clustering techniques, chosen to represent distant node relationships? Would clustering provide richer information than distance measures alone?
3. Can the authors provide a more detailed explanation of the experimental setup and discuss specific findings, such as the T-H2GCN performance, to better validate their approach?

---

### Official Review · Reviewer_7WKa · 2024-11-04

**Soundness:** 2
**Presentation:** 2
**Contribution:** 1
**Rating:** 3
**Confidence:** 4

**Summary:**

The paper presents WISE-GNN, a framework designed to address the limitations of GNNs in capturing long-range dependencies, particularly in heterophilic graphs, by incorporating Wise-embeddings and a topological module.

**Strengths:**

The paper is generally clear.

 Promising performances have been achieved.

**Weaknesses:**

1. The novelty is limited considering that class landmark (or prototypes) has been extensively used in node representation learning and clustering in the literature.
2. The wise embedding seem to be handcrafted. This could be time-consuming. The authors did not discuss the runtime of the method.
3. It is unclear why wise embedding is used for GNN input while the betti vector is processed through MLP.
4. The Chameleon and Squirrel datasets used in the study are known to be problematic[1]. It is recommended to use the filtered versions of these datasets to ensure more reliable results.
5. The heterophilous datasets used for evaluation are all relatively small. To validate the claims more robustly, it is essential to conduct experiments on large-scale heterophilous datasets[1][2].
6. More recent heterophilous baselines are needed for performance comparison.
7. Reproducibility is poor. No code released, and no hyperparameters specified.


References

[1] Platonov, O., Kuznedelev, D., Diskin, M., Babenko, A., & Prokhorenkova, L. (2023). A critical look at the evaluation of GNNs under heterophily: Are we really making progress?. arXiv preprint arXiv:2302.11640.

[2] Derek Lim, Felix Hohne, Xiuyu Li, Sijia Linda Huang, Vaishnavi Gupta, Omkar Bhalerao, and Ser Nam Lim. Large scale learning on non-homophilous graphs: New benchmarks and strong simple methods. Advances in Neural Information Processing Systems, 34, 2021.

**Questions:**

Please see weaknesses.

**Details Of Ethics Concerns:**

I suspect that the submission currently under review may overlap significantly with a paper submitted to AAAI, titled “ClassContrast: Bridging the Spatial and Contextual Gaps for Node Representations”.https://arxiv.org/pdf/2410.02158

Upon comparing the two papers, it appears that approximately 50% of the content overlaps. Specifically, both papers propose the use of class-wise landmarks and leverage the distance between individual nodes and these landmarks as additional node embeddings. This concept seems to be a major contribution in both works, raising the concern of a potential dual submission.

---

### Official Review · Reviewer_wrKp · 2024-11-04

**Soundness:** 2
**Presentation:** 1
**Contribution:** 1
**Rating:** 1
**Confidence:** 5

**Summary:**

This paper aims to address the limitations of graph neural networks to improve the performance of graph neural network on heterophilic and homophilic graphs. Specifically, a wise embedding is proposed to encode the topological information to benefit the classification with different GNN architectures. Experiments are conducted on several small datasets.

**Strengths:**

1. The background is introduced clearly.
2. The results of visualization are interesting.

**Weaknesses:**

1. The motivation of this paper is very unclear. In the abstract and introduction, the authors mention that they want to address the issues of GNNs in oversmoothing, oversquashing, and underreaching. However, what are these problems are not introduced in detail. In this work, a method of topology embedding is proposed. It's very unclear why this can solve these many problems of GNNs. Moreover, in the experiments, the authors majorly evaluate the performance on heterophilic graphs, which make the objective of this work more confusing.

2. The technical contributions are limited. The core idea of this method is to learn a topology embedding to benefit the classification with GNNs. Similar ideas have already been well explored. For example, LinkX utilizes an MLP to learn the topology embedding from the adjacency matrix. Position embeddings are also clearly investigated for graph transformer [1].

3. The experiments are not convincing. For example, in table 2, the GCN only has 59% accuracy, which is in conflict with the widely reported results (around 72%). Hence, the improvements are not convincing. In addition, experiments are only conducted on several small datasets which only contain several thousand nodes.

4. The paper is poorly written. The details of building the wise embedding is difficult to follow. Why this method can be effective for heterophilic graphs is not discussed. The paper is not cited is a right way, which makes the paper very difficult to follow.

[1] Ying, Chengxuan, et al. "Do transformers really perform badly for graph representation?." Advances in neural information processing systems 34 (2021): 28877-28888.

**Questions:**

Please refer to the weaknesses.

---

### Note · Authors · 2024-11-15

**Comment:**

We thank the reviewers for their time and valuable feedback. We will incorporate your suggestions and comments to improve the paper and submit it to another venue.

**Withdrawal Confirmation:**

I have read and agree with the venue's withdrawal policy on behalf of myself and my co-authors.